# Validation of the “Rome” Classification for Squamous Cell Carcinoma of the Nasal Vestibule

**DOI:** 10.3390/cancers16010037

**Published:** 2023-12-20

**Authors:** Willem Frederik Julius Scheurleer, Mischa de Ridder, Luca Tagliaferri, Claudia Crescio, Claudio Parrilla, Gian Carlo Mattiucci, Bruno Fionda, Alberto Deganello, Jacopo Galli, Remco de Bree, Johannes A. Rijken, Francesco Bussu

**Affiliations:** 1Department of Head and Neck Surgical Oncology, University Medical Center Utrecht, Heidelberglaan 100, 3584 CX Utrecht, The Netherlands; r.debree@umcutrecht.nl (R.d.B.); j.a.rijken-3@umcutrecht.nl (J.A.R.); 2Department of Radiation Oncology, University Medical Center Utrecht, Heidelberglaan 100, 3584 CX Utrecht, The Netherlands; m.deridder-5@umcutrecht.nl; 3Radiation Oncology Division, Department of Diagnostic Imaging, Radiation Oncology and Hematology, Fondazione Policlinico Universitario “A. Gemelli” IRCCS, Largo Agostino Gemelli 8, 00168 Rome, Italy; luca.tagliaferri@policlinicogemelli.it (L.T.); bruno.fionda@policlinicogemelli.it (B.F.); 4Otolaryngology Division, Azienda Ospedaliero Universitaria Sassari, Viale S. Pietro 43, 07100 Sassari, Italy; claudia.crescio@aouss.it (C.C.); fbussu@uniss.it (F.B.); 5Department of Medicine, Surgery and Pharmacy, University of Sassari, Viale S. Pietro 43, 07100 Sassari, Italy; 6Otolaryngology Division, Department of Neurosciences, Sensory Organs and Thorax, Fondazione Policlinico Universitario “A. Gemelli” IRCCS, Largo Agostino Gemelli 8, 00168 Rome, Italy; claudio.parrilla@policlinicogemelli.it (C.P.); jacopo.galli@policlinicogemelli.it (J.G.); 7Department of Radiation Oncology, Mater Olbia Hospital, SS 125 Orientale Sarda, 07026 Olbia, Italy; giancarlo.mattiucci@materolbia.com; 8Department of Otolaryngology-Head and Neck Surgery, IRCCS Istituto Nazionale dei Tumori (INT), 20133 Milan, Italy; alberto.deganello@istitutotumori.mi.it

**Keywords:** nasal vestibule, squamous cell carcinoma, sinonasal cancer, staging

## Abstract

**Simple Summary:**

Nasal vestibule cancer is considered to be a rare form of cancer. There are several different staging systems in place for categorizing these tumors, which can result in inconsistencies and unreliable data due to variations in registration. The “Rome” classification is the most recent addition. This study aimed to assess the effectiveness of this new staging system and to compare it to the UICC/AJCC system. One hundred and forty-nine patients with a squamous cell carcinoma of the nasal vestibule were included. There was a significant association between an increased disease stage as staged per the Rome classification and decreased survival. This association persisted when correcting for various covariates (i.e., age at the time of diagnosis, sex, the presence of lymph node metastases, and treatment modality) in multivariable analysis. The Rome classification appears to be capable of adequately categorizing and stratifying patients for different outcome measures. Nevertheless, additional research with a larger number of patients is required before any definitive conclusions can be drawn.

**Abstract:**

Squamous cell carcinoma of the nasal vestibule is considered a rare malignancy that differs from other sinonasal malignancies in many respects. Four staging systems currently exist for this disease, the most recent addition being the “Rome” classification. This study assesses the use of this new classification and its prognostic value regarding various outcome measures. A retrospective multicenter cohort study of patients with a primary squamous cell carcinoma of the nasal vestibule who were treated in three tertiary head and neck oncology referral centers was conducted. A total of 149 patients were included. The median follow-up duration was 27 months. Five-year locoregional control (LRC), disease-specific survival (DSS), and overall survival (OS) were 81.6%, 90.1, and 62.5% respectively. A statistically significant association was observed between the Rome classification and all survival outcomes in both univariable and multivariable analyses. Moreover, it appeared to perform better than the Union for International Cancer Control TNM classification for tumors of the nasal cavity and paranasal sinuses. The new Rome classification can be used effectively and is associated with LRC, DSS, and OS. However, it requires further validation in a larger (prospective) study population.

## 1. Introduction

Squamous cell carcinoma of the nasal vestibule (SCCNV) is a peculiar disease. It is categorized amongst malignancies of the nasal cavity and paranasal sinuses, even though it differs from tumors of the nasal cavity proper in many respects. Yet, it lacks a designated topography code and is thus not registered as a distinct entity. These tumors are considered to be rare and reportedly comprise less than one percent of head and neck cancers, which may be an underestimation as a result of poor registration [1]. Due to its prominent location in the midface, SCCNV can be diagnosed at an early stage, and lymph node metastases are rarely present at first presentation [2]. However, this is dependent on both patients’ as well as physicians’ awareness [3]. Surgery, with or without adjuvant radiotherapy, is the foundation of treatment for (locally) advanced SCCNV, in particular in the case of bone involvement [4,5,6,7,8,9,10,11]. Both surgery and external beam radiotherapy (EBRT) provide great oncological outcomes [4,8,9,11,12,13,14,15,16,17,18]. Brachytherapy (also known as interventional radiotherapy), however, has increasingly become the treatment modality of choice for early-stage SCCNV, as it produces superior survival, aesthetic, and functional outcomes compared to other options [6,8,16,19,20,21,22,23,24]; this is mainly because surgery may lead to deformation and require additional reconstruction [4,10,11,18,23,25]. Meanwhile, systemic therapy (i.e., chemotherapy, immunotherapy, or targeted therapy) is mostly reserved for niche cases (e.g., distant metastases or inoperable disease) and is generally not considered standard treatment [26]. 

Three staging systems have been adopted in clinical practice to varying degrees. These are the Union for International Cancer Control (UICC) TNM classification (8th ed.) for tumors of the nasal cavity and paranasal sinuses, the UICC TNM classification (8th ed.) for non-melanoma skin cancer of the head and neck, and the Wang classification [27,28]. These all have their perks and shortcomings, and no consensus has been reached as to which system performs best, resulting in inconsistent registration and unnecessary heterogeneity of data [17,19,29,30]. Major downsides of the available staging systems are their insufficient practical applicability and the lack of integration of modern high-resolution imaging [31]. The new classification by Bussu et al. has recently been introduced in an attempt to replace the aforementioned options by focusing on clinical usability, the site’s unique anatomy, and the typical patterns of disease spread [32]. A universally adopted, properly designed staging system can help create uniformity in staging, registration, and, eventually, treatment [31]. Following the prior in-depth technical evaluation of the four existing staging systems for SCCNV, this study assesses the use of the novel Rome classification and its association with various survival outcomes. Additionally, the aim is to compare its performance to the UICC TNM classification for tumors of the nasal cavity and paranasal sinuses.

## 2. Materials and Methods

### 2.1. Study Population

Cases were included from three tertiary head and neck oncology referral centers in Italy (Fondazione Policlinico Universitario “A. Gemelli” IRCCS and Azienda Ospedaliera Universitaria di Sassari) and The Netherlands (University Medical Center Utrecht). All consecutive adult patients with a histopathologically confirmed primary SCCNV who had been diagnosed between August 2004 and August 2023 were eligible for inclusion. Patients diagnosed with other histological subtypes (e.g., basal cell carcinoma, Merkel cell carcinoma, melanoma) were excluded. Part of this cohort has been described in a previous publication [31]. Each patient underwent comprehensive clinical assessment by a head and neck surgeon and radiation oncologist. Imaging studies consisted of magnetic resonance imaging (MRI) or computed tomography (CT) of the head and neck, alongside chest X-ray/CT and neck ultrasound, with or without fine-needle aspiration cytology of suspicious lymph nodes. All patients were discussed in the local multidisciplinary tumor board prior to treatment selection. Selection of treatment modalities occurred on a case-by-case basis and in consultation with patients.

By assessing clinical data, all tumors were restaged in accordance with the UICC TNM classification for tumors of the nasal cavity and paranasal sinuses (8th ed.), as well as the new classification per Bussu et al. (henceforth referred to as the Rome classification) (Table 1) [32]. Follow-up duration was determined from the date of histopathological confirmation of diagnosis to the most recent moment of clinical follow-up or the date of death. This study was approved by the University Medical Center Utrecht medical ethics committee (22-859). For this type of study, formal consent was not required. All participating centers complied with local medical ethics committee requirements.

### 2.2. Statistical Analysis

SPSS version 29.0 was utilized to conduct all statistical analyses. Normality was assessed through Kolmogorov–Smirnov tests and Q-Q plots. Patient characteristics were expressed as either means with standard deviations (SD) for normally distributed variables or medians with the 25th and 75th percentiles for variables that did not conform to normal distribution. Locoregional control (LRC) was measured from the date of diagnosis until the date of confirmation of local and/or regional recurrence. Disease-specific survival (DSS) was measured from the date of diagnosis until the date of death due to SCCNV. Overall survival (OS) was measured from the date of diagnosis until the date of death due to any cause. Survival rates were calculated using the Kaplan–Meier method [33]. Univariable and multivariable analyses of association were performed using the Cox proportional hazards model. The proportional hazards assumption was visually assessed for categorical variables. For continuous variables, an interaction with time was examined. All analyzed variables met the proportional hazard assumption. Treatment modalities were grouped in order to ensure adequate sample size for each of the categories. Patients who did not undergo treatment or received best supportive care were excluded from all analyses relating to LRC (i.e., Kaplan–Meier estimates and univariable Cox proportional hazard analysis). Multivariable analyses could not be performed for LRC and DSS due to the infrequent occurrence of events. Age at the time of diagnosis and sex were included in multivariable analysis for OS regardless of their statistical significance. Other variables with a *p*-value < 0.10 in univariable analysis were introduced in multivariable analysis and were subsequently eliminated in a stepwise-backward manner. The Rome classification and UICC TNM classification for tumors of the nasal cavity and paranasal sinuses were not entered into the same multivariable model. Probability (*p*) values < 0.05 were deemed to be statistically significant for all tests.

## 3. Results

A total of 149 patients were identified and included in this multicenter cohort. Their clinical characteristics are disclosed in Table 2. 

Imaging protocols differed across participating centers. Out of 149 patients, 107 (71.8%) underwent an MRI of the head and neck, while 87 (58.4%) underwent a CT scan. Fifty-three (35.6%) patients underwent both. For the purpose of further staging, a majority of patients underwent a neck ultrasound (83.2%) and/or chest X-ray/CT. Notably, 25 (16.8%) patients underwent a positron emission tomography (PET)/CT. Two patients (1.3%), both of whom refused treatment, did not undergo any imaging.

Out of 149 patients, more than half (50.3%) were staged T1 according to the Rome classification. Twenty-seven (18.1%) patients were staged T2a due to invasion of (sub)cutaneous tissues, and twenty-four (16.1%) displayed cartilage invasion and were thus staged T2b. Fifteen (10.1%) patients were staged T3 because of tumor extension beyond the pyriform aperture, and eight (5.4%) were staged T4a on account of bony invasion. Nine patients (6.0%) presented with lymph node metastases, whilst none had developed distant metastases at first clinical presentation.

All but two (98.7%) patients were treated with curative intent. In this cohort of 149 patients, 113 (85.6%) received brachytherapy and 16 (12.1%) received EBRT, whilst 3 (2.0%) received a combination of both. Fourteen (9.4%) patients underwent surgery, five (3.4%) of whom received adjuvant EBRT. One (0.7%) of these patients also received neoadjuvant immune therapy (nivolumab, two cycles) for a cT4aN2cM0 tumor, and one (0.7%) patient was treated with chemoradiotherapy (cisplatin/etoposide) for a cT4aN0M0 tumor with bony invasion of the maxilla because of a synchronous sinonasal neuroendocrine carcinoma of the maxillary sinus. Two (1.3%) patients were treated with palliative intent and received best supportive care, one of whom was lost to follow-up. 

Out of 147 patients who were treated with curative intent, 23 (15.6%) developed recurrent disease. Thirteen (8.8%) patients had a local recurrence, seven (4.8%) had a regional recurrence, and one (0.7%) developed a distant recurrence, whilst two (1.4%) patients developed simultaneous regional and distant recurrences. The 5-year LRC was 81.6%, and all recurrences occurred within 3 years of follow-up. At the most recent moment of follow-up, out of 149 patients, 98 (65.8%) had no evidence of disease, 40 (26.8%) died of intercurrent disease, and 10 (6.7%) died as a result of SCCNV. The estimated 5-year DSS for the entire population was 90.1%, whilst the 5-year OS was 62.5%. Kaplan–Meier estimates for LRC, DSS, and OS stratified for the cT stages of the Rome classification are shown in Figure 1A−C. When excluding N+ patients, the 5-year LRC, DSS, and OS were 84.1%, 92.3%, and 63.6%, respectively (Figure 1D–F). There was a statistically significant difference in survival between different stages for all three outcome measures. The same could be observed for the UICC TNM classification for tumors of the nasal cavity and paranasal sinuses. However, a higher T-stage did not always correspond with poorer outcomes. Patients with a cT2 tumor had worse outcomes compared to patients with cT4a tumors (Figure 1G–I).

Univariable and multivariable Cox proportional hazard analyses are displayed in Table 3. There was a statistically significant association between LRC, DSS, OS, and the Rome classification and cN stage in univariable analysis. The treatment modalities employed for treating the primary tumor were associated with LRC and OS but not DSS. The statistically significant association between age at the time of diagnosis, Rome classification, and cN stage and OS persisted in multivariable analysis. 

The UICC TNM classification for tumors of the nasal cavity and paranasal sinuses was also significantly associated with LRC, DSS, and OS. However, notably, the hazard ratio (HR) for patients with T2 disease was higher compared to the HR for patients with a T4a tumor (Table 3). The HR for T3 disease was 0.00 for DSS and OS due to the absence of events. Similar to the Rome classification, a statistically significant association between age at the time of diagnosis, cN stage, the UICC TNM classification for tumors of the nasal cavity and paranasal sinuses and OS persisted in multivariable analysis.

## 4. Discussion

In this retrospective multicenter study, a cohort of 149 patients who were treated for a primary SCCNV is described. The 5-year LRC, DSS, and OS were 81.6%, 90.1%, and 62.5%, respectively. This is largely in line with previous literature [6,11,16,19,21,24,29]. Yet, side-by-side comparison to other studies is difficult and impractical because of the variation in treatment modalities between cohorts, as well as the differences in treatment protocols between participating centers.

A disease classification should enable the categorization of patients based on well-defined criteria. The distribution within these categories should be such that a physician can not only determine the most appropriate treatment approach but also determine the prognosis for an individual patient based on their disease stage. As for the staging of SCCNV, there is general consensus regarding the staging of lymph node metastases, which follows that of other head and neck subsites (with the exception of HPV-related oropharyngeal squamous cell carcinoma as well as nasopharyngeal carcinoma). However, the classification of the primary tumor is controversial. Three different systems have been used: the UICC TNM classification (8th ed.) for tumors of the nasal cavity and paranasal sinuses, the UICC TNM classification (8th ed.) for non-melanoma skin cancer of the head and neck (which is by definition inappropriate according to WHO, as the nasal vestibule is part of the nasal cavity) and the Wang classification [27,28]. Their individual advantages and disadvantages have already been discussed at length [31]. The Rome classification was proposed with the aim of being easier to use in clinical practice through focusing on anatomical landmarks and by allowing the integration of high-resolution imaging techniques. It incorporates skin, cartilage, and bone involvement as clear determinants and was conceived with the aim of being integrated in the UICC TNM staging system. It has previously been suggested that the Rome classification leads to a more balanced allocation amongst stages, but its prognostic value requires further evaluation. The results from this study show an association between the Rome classification and all investigated survival outcomes, both in univariable and multivariable analyses. Moreover, there was a clear increase in hazard with increasing disease stage for all reported outcomes. This had not been researched previously. This study also reports a statistically significant difference in outcome between different stages, showing that the Rome classification is capable of adequately discriminating between patients at baseline.

The UICC TNM classification for tumors of the nasal cavity and paranasal sinuses was also significantly associated with all survival outcomes in this cohort. In this cohort, patients with cT2 disease had worse outcomes compared to those with cT4a tumors. Yet, logic would dictate that high disease stage would lead to poor outcomes, which is not the case. Unfortunately, outcomes for patients with cT3 disease do not allow for clear interpretation. This is the result of small sample size (n = 2) and the subsequent non-occurrence of events. Inclusion of such a category in analyses is not ideal, but nevertheless essential when comparing different classifications. As such, this imbalanced allocation amongst stages points out practical problems with the UICC TNM classification for tumors of the nasal cavity and paranasal sinuses when applied to SCCNV. Clearly, the definitions are not tailored to this subset of tumors. This further stresses the inadequacies of the current UICC system when it comes to both the actual staging of patients with SCCNV, as well as stratification.

Several limitations ought to be taken into consideration. Analyses were impaired by power issues due to the relatively small sample size and the infrequent occurrence of survival endpoints. As a result, multivariable Cox proportional hazard analysis for LRC and DSS, as well as additional subgroup analyses, could not be performed. A larger (prospective) cohort should ideally be established for the sake of definitive validation of the Rome classification and to establish whether or not it is superior to the other options. Ideally, such a cohort should consist of patients who have been treated with a variety of treatment modalities, including surgery, brachytherapy, and EBRT. This should provide additional information on whether the core premise of the Rome classification (i.e., involvement of specific anatomical structures) actually allows for better stratification, treatment selection, and treatment planning. Future research should also investigate whether other factors, such as tumor diameter or the subsite of origin (e.g., vestibular septum, dome, or floor), provide additional prognostic value. If so, incorporation of such factors ought to be considered. The subsequent step should be to set up a specific ICD topography code for the nasal vestibule in order to clearly distinguish it from the ethmoid sinus and the nasal cavity proper as a third site (alongside the maxillary sinus) for the staging of sinonasal malignancies. Over time, through accurate registration, this should lead to a better understanding of the incidence of SCCNV. Here, both rapid adoption and patience are crucial, as this progress requires time. A secondary benefit would be the improvement of the registration of cancer of the nasal cavity property/ethmoid sinus because these numbers are inflated due to the inclusion of SCCNV.

## 5. Conclusions

The necessity for a dedicated classification and topography code for the nasal vestibule has previously been established. The inadequacy of the current UICC system for the purpose of staging SCCNV is displayed in the present series. The new Rome classification can be used effectively and may be a good prognostic indicator for LRC, DSS, and OS. However, additional validation in a larger cohort is required before it can be widely adopted.

## Figures and Tables

**Figure 1 cancers-16-00037-f001:**
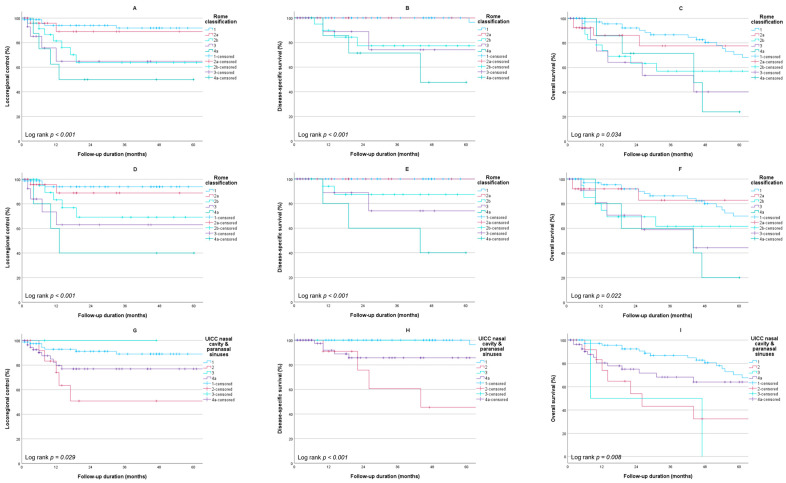
Kaplan–Meier survival estimates stratified per stage. **Rome classification for the entire cohort**: (**A**) locoregional control (n = 147); (**B**) disease-specific survival (n = 149); (**C**) overall survival (n = 149); **Rome classification for the cohort excluding N+ patients**: (**D**) locoregional control (n = 138); (**E**) disease-specific survival (n = 140); (**F**) overall survival (n = 140); **UICC nasal cavity and paranasal sinuses for the entire cohort**: (**G**) locoregional control (n = 147); (**H**) disease-specific survival (n = 149); (**I**) overall survival (n = 149).

**Table 1 cancers-16-00037-t001:** Description of the UICC TNM classification for tumors of the nasal cavity and paranasal sinuses and the Rome classification.

Stage	UICC TNM Nasal Cavity and Paranasal Sinuses	Rome Classification
**T1**	Tumor restricted to any one subsite, with or without bony invasion.	Tumor limited to the internal lining of the nasal vestibule (skin and/or mucosa).
**T2(a)**	Tumor invading two subsites in a single region or extending to involve an adjacent region within the nasoethmoidal complex, with or without bony invasion.	Tumor invading superficial structures (cutis, subcutis) beyond the nasal cavity, in particular the external surface of the nose, the nasolabial fold, philtrum, or upper lip, without invasion of cartilage, bone, or structures beyond the plane of the pyriform aperture.
**T2b**		Tumor invading cartilage (quadrangular, triangular, alar) without invasion of bony structures or structures beyond the plane of the pyriform aperture.
**T3**	Tumor extends to invade the medial wall or floor of the orbit, maxillary sinus, palate, or cribriform plate.	Tumor extends posteriorly beyond the plane of the pyriform aperture, with or without cartilage invasion, but without bone invasion.
**T4a**	Moderately advanced local disease. Tumor invades any of the following: anterior orbital contents, skin of nose or cheek, minimal extension to anterior cranial fossa, pterygoid plates, sphenoid or frontal sinuses.	Tumor invades bony structures (e.g., hard palate, nasal bones, frontal process of the maxilla, ethmoid, or orbit).
**T4b**	Very advanced local disease. Tumor invades any of the following: orbital apex, dura, brain, middle cranial fossa, cranial nerves other than (V2), nasopharynx, or clivus.	Tumor invades any of the following: orbital apex, dura, brain, middle cranial fossa, cranial nerves other than (V2), nasopharynx, or clivus.

**Table 2 cancers-16-00037-t002:** Patient characteristics for 149 patients with a primary squamous cell carcinoma of the nasal vestibule.

**Sex**	**N**	**%**
Male	92	61.7
Female	57	38.3
**Age at the time of diagnosis**	**N**	**SD**
Mean (years)	68.6	11.6
**Imaging studies**	**N**	**%**
MRI-head/neck	107	71.8
CT-head/neck	87	58.4
Neck ultrasound	124	83.2
Chest X-ray/CT	131	87.9
PET/CT	25	16.8
**cT stage**	**N**	**%**
Rome classification		
T1	75	50.3
T2a	27	18.1
T2b	24	16.1
T3	15	10.1
T4a	8	5.4
UICC nasal cavity and paranasal sinuses		
T1	77	51.7
T2	14	9.4
T3	2	1.3
T4a	56	37.6
**cN stage**	**N**	**%**
N0	140	94.0
N+	9	6.0
**Tumor diameter**	**N**	**%**
<15 mm	47	31.5
≥15 mm	59	39.6
Unknown	43	28.9
**Primary tumor treatment modality**	**N**	**%**
Brachytherapy	113	75.8
EBRT	16	10.7
Brachytherapy + EBRT	3	2.0
Surgery (+EBRT)	14	9.4
Chemoradiotherapy	1	0.7
None/Best supportive care	2	1.3
**Neck treatment modality**	**N**	**%**
EBRT	3	2.0
Neck dissection	8	5.4
Neck dissection + EBRT	1	0.7
None	137	91.9
**Follow-up**	**N**	**p25–p75**
Median duration (months)	27	9.5–62.5
**Outcome**	**N**	**%**
NED	98	65.8
DID	40	26.8
DOD	10	6.7
LTF	1	0.7

SD = standard deviation; EBRT = external beam radiotherapy; p25–p75 = 25th percentile–75th percentile; NED = no evidence of disease; DID = died of intercurrent disease; DOD = died of disease; LTF = lost to follow-up.

**Table 3 cancers-16-00037-t003:** Univariable and multivariable Cox proportional hazard analysis for locoregional control, disease-specific survival, and overall survival.

	LRC ^u^		DSS ^u^		OS ^u^		OS ^m^	
	HR	*p*	HR	*p*	HR	*p*	HR	*p*
**Age (per year)**	1.01	0.497	0.98	0.373	1.05	**<0.001**	1.06 *	**<0.001 ***
**Sex**		0.467		0.236		0.154		ns
Male	ref		ref		ref		.	.
Female	0.72		0.39		0.64		.	
**cT stage**								
Rome classification		**0.005**		**0.040**		**0.048**		**0.044 ***
T1	ref		ref		ref		ref	
T2a	1.53		0.00		0.95		1.19	
T2b	5.09		18.9		2.02		2.12	
T3	6.18		19.5		2.97		3.29	
T4a	8.74		43.6		2.94		3.29	
UICC nasal cavity and paranasal sinuses		**0.042**		**0.019**		**0.016**		**0.002**
T1	ref		ref		ref		ref	
T2	5.23		32.2		2.84		1.40	
T3	0.00		0.00		6.20		2.22	
T4a	2.51		10.7		1.43		0.52	
**cN stage**		**0.018**		**0.002**		**0.024**		**0.008 ***
N0	ref		ref		ref		ref	
N+	3.71		8.81		2.96		4.37	
**Primary tumor treatment modality**		**0.012**		0.610		**0.018**		ns
Surgery (+EBRT)	ref		ref		ref		.	
Brachytherapy (+EBRT)	0.52		0.41		0.72		.	
EBRT (+chemotherapy)	2.14		1.00		1.84		.	
None	.		0.00		6.42		.	

Age was included as a continuous variable. LRC = locoregional control; DSS = disease-specific survival; OS = overall survival; u = univariable; m = multivariable; HR = hazard ratio; ns = not significant. The brachytherapy group includes patients who received both brachytherapy and EBRT. The EBRT group includes patients who received both EBRT and chemotherapy. *p*-values < 0.05 are displayed in bold. Variables with a *p*-value < 0.10 were introduced in multivariable analysis and eliminated in a stepwise-backward manner. * The Rome classification and UICC TNM classification for tumors of the nasal cavity and paranasal sinuses were not entered into the same multivariable model. Values for all other variables correspond to the multivariable model, including the Rome classification.

## Data Availability

The data presented in this study are available on request from the corresponding author. The data are not publicly available due to privacy-related restrictions.

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
