# Peer review of "Validation of the “Rome” Classification for Squamous Cell Carcinoma of the Nasal Vestibule"

_cancers, 2023, doi:10.3390/cancers16010037_

Round 1

Reviewer 1 Report

Comments and Suggestions for Authors

The authors aimed at investigating the prognostic value of Bussu et al. classification for nasal vestibule cancer or “Rome” classification as called by the authors. The subject is interesting and stimulating since there is a need to find an homogenous and widely accepted classification for this subsite of sinonasal cancer.

Cohort of patients is large, but median follow-up is short (27 months) and survival outcome (OS=62.5%) seems low considering that more than half of patients are staged as a T1.

A limit of the study is that the present article aims to validate the Rome classification on series of patients treated mostly with radiotherapy protocols (only 14/149 surgical cases and only one treated with systemic treatment). Authors should comment extensively on this and the Rome classification should be validated also in surgical series to prove to be a valid or superior staging system in comparison with the UICC one. 

Abstract.

Line 46-47: Moreover,  it seemed to achieve? superior performance compared to the Union for International Cancer Control TNM- classification for tumors of the nasal cavity and paranasal sinuses 

Introduction.

Line 65-67: authors state that BT is the modality of choice for early stage SCCNV but this is what they stated in groups previous papers and has not been validated from external groups yet (it should be stated and discussed, also considering the low OS that was reported already in papers from the same group).

Methods section.

The methodology could be improved.

Line 95-96: can you better explain the meaning of this sentence? Or it is a misprint?

Line 104-105 why not starting follow-up at the end of primary treatment?

Therapeutic decision-making process and indication could be clearer.
Line 115-118 follow-up should start later at the end of primary treatment.

Results / Discussion:

Median follow-up is very short to estimate 5y main outcomes and to state conclusions.

Very high DSS and low OS, despite high percentage of T1, authors should investigate and describe this aspect better.

Figure 1 is too tiny, readers can barely see KM curves and it is too hard to understand differences and colors.

Table 2 should fit all in the same page. And in Table 2 percentages are not correct in "primary tumor treatment modality" (e.g. 85.6% should be 113x100/149=75.8% and so on).

Line 232-233 it’s a paper from the same group and using the verb "established" could be considered excessive to drive reader to such a strong conclusion.

Looking at KM curves in DSS and OS there is not a big difference in outcomes between T1 and T2a according to Rome Class. Merging categories T1 and T2a (leaving the T2b as simple T2) should be considered?

Reviewer 2 Report

Comments and Suggestions for Authors

Thank you for submitting your manuscript to our journal. I enjoyed reading your work and I understand its rationale and the effort that went into the project. Please see below for my comments.

1) One of the premises of the paper is that SCCNV is poorly described and stratified by the current classification systems. There is merit to this- anatomically it belongs to the nasal cavity but histologically it's more consistent with non-melanoma skin cancer. The major advantage with the Rome classification is that it is specific for SCCNV, and this is apparent to the readers who have not seen it before. I would urge the authors to include the current staging system (UICC-TNM) to emphasize the difference.

2) While the Rome classification benefits from disease specific description, it is not without flaws. In the end, patients want to be properly staged for prognostic and survival outcomes purposes, but surgeons also want to know the anatomic "extensiveness" of the tumor for surgical planning. I think the Rome system does well delineating the differences between stages (the limit of pyriform aperture for example), however, the behavior the tumor is also dependent on the site of origin and may not completely reflect the tumor biology. For example, if the tumor originates from a midline structure, such as the cutaneous columella, by proximity it may involve the cartilaginous septum (upstage to T2b)  or the alveolus/hard palate (upstage to T4a) before it extends beyond the pyriform aperture. I think there is some merit to the TNM staging system where a T1 lesion is limited to the single site but allows for bony invasion. Nevertheless, this could be addressed in the discussion section in comparing the two classification systems and the authors can highlight it in future directions they want to take.

Reviewer 3 Report

Comments and Suggestions for Authors

Congratulations to the AA for the excellent paper; it  provides undoubted innovations to the classification system of carcinoma of the nasal vestibule on the basis of a retrospective study which includes a sufficiently large series of cases
